

# Impacts of short-term mitigation measures on PM2.5 and radiative effects: a case study from a regional background site near Beijing, China

Qiyuan Wang[1*], Suixin Liu[1], Nan Li[2], Wenting Dai[1], Yunfei Wu[3], Jie Tian[4], Yaqing Zhou[1], Meng Wang[1], Steven Sai Hang Ho[1], Yang Chen[5], Renjian Zhang[3], Shuyu Zhao[1], Chongshu Zhu[1], Yongming Han[1,6], Xuexi Tie[1], Junji Cao[1,7*]

[1]Key Laboratory of Aerosol Chemistry and Physics, State Key Laboratory of Loess and Quaternary Geology, Institute of Earth Environment, Chinese Academy of Sciences, Xi'an, 710061, China.

[2]School of Environmental Science and Engineering, Nanjing University of Information Science & Technology, Nanjing, 210044, China

[3]Key Laboratory of Regional Climate-Environment Research for Temperate East Asia, Institute of Atmospheric Physics, Chinese Academy of Sciences, Beijing, 100029, China.

[4]Department of Environmental Science and Engineering, School of Energy and Power Engineering, Xi'an Jiaotong University, Xi'an 710049, China.

[5]Chongqing Institute of Green and Intelligent Technology, Chinese Academy of Sciences, Chongqing 400714, China.

[6]School of Human Settlements and Civil Engineering, Xi'an Jiaotong University, Xi'an, 710049, China.

[7]Institute of Global Environmental Change, Xi'an Jiaotong University, Xi'an, 710049, China.

*Correspondence to*: Junji Cao (cao@loess.llqg.ac.cn) and Qiyuan Wang (wangqy@ieecas.cn)

**Abstract.** An intensive measurement campaign was conducted in a regional background site near Beijing during the 19th National Congress of the Communist Party of China (NCCPC) to investigate the effectiveness of short-term mitigation measures on PM2.5 and aerosol direct radiative forcing (DRF). Average mass concentration of PM2.5 and its major chemical composition are decreased by 20.6 – 43.1% during the NCCPC control period compared with the non-control period. When considering days with the stable meteorological conditions, larger reduction of PM2.5 is found compared with that for all days. Further, a positive matrix factorization receptor model shows that the mass concentrations of PM2.5 from traffic-related emissions, biomass burning, industry processes, and mineral dust are reduced by 38.5 – 77.8% during the NCCPC control period compared with the non-control period. However, there is no significant difference in PM2.5 from coal burning between these two periods, and an increasing trend of PM2.5 mass from secondary inorganic aerosol is found during the NCCPC control period. Two pollution episodes were occurred subsequently after the NCCPC control period. One is dominated by secondary inorganic aerosol, and the WRF-Chem model shows that the Beijing-Tianjin-Hebei (BTH) region contributes 73.6% of PM2.5 mass; the other is mainly caused by biomass burning, and the BTH region contributes 46.9% of PM2.5 mass. Calculations based on a revised IMPROVE method show that organic matter (OM) is the largest contributor to the light extinction coefficient ($b_{ext}$) during the non-control period while NH4NO3 is the dominant contributor during the NCCPC control period. The Tropospheric Ultraviolet and Visible radiation model reveals that the average DRF values at the Earth's surface are -14.0 and -19.3 W m$^{-2}$ during the NCCPC control and non-control periods, respectively, and the reduction ratios of DRF due to the



decrease in PM$_{2.5}$ components vary from 22.7 – 46.7% during the NCCPC control period. Our study would further provide
valuable information and dataset to help controlling the air pollution and alleviating the cooling effects of aerosols at the
surface in Beijing.

**1 Introduction**

High loadings of fine particulate matter (PM$_{2.5}$, particulate matter with an aerodynamic diameter $\leq$ 2.5 μm) can strongly
deteriorate air quality (Pui et al., 2014; Tao et al., 2017), reduce atmospheric visibility (Watson, et al., 2002; Cao et al., 2012),
and cause adverse effects to human health (Feng et al., 2016; Xie et al., 2016). Moreover, PM$_{2.5}$ also directly or indirectly
affect climate and ecosystem (Lecoeur et al., 2014; Tie et al., 2016). With the rapid increases in economic growth,
industrialization, and urbanization in the past two decades, Beijing has been experienced serious PM$_{2.5}$ pollution, especially in
winter (e.g., Zhang et al., 2013; Elser et al., 2016; Wang et al., 2016a; Zhong et al., 2018). Since the Chinese government
promulgated the new National Ambient Air Quality Standards (NAAQS, GB3095–2012) for PM$_{2.5}$ in 2012, a series of emission
control strategies have been performed in Beijing and its surrounding areas to alleviate the serious air pollution problems, e.g.
installing strong desulphurization system in coal-fired power plants, eliminating high emission motor vehicles, and promoting
natural gas instead of coal in rural areas. According to the data issued by the China Environmental State Bulletin
(www.zhb.gov.cn/hjzl/zghjzkgb/lnzghjzkgb, in Chinese), although the annual level of PM$_{2.5}$ during 2013 – 2016 in Beijing
shows a decreasing trend (r = 0.98 and slope = -5.3 μg m$^{-3}$ year$^{-1}$), there are still 45.9% of days in 2016 suffering from different
degrees of pollution.
The causes of air pollution in Beijing are complicated due to the complex chemical composition in PM$_{2.5}$ and their different
formation processes. For example, Elser et al. (2016) have reported that organic aerosol (OA) is the largest contributor to PM$_{2.5}$
mass during the extreme haze periods in Beijing, and the primary aerosol from coal combustion (46.8%) is the dominant
contributor to OA, followed by the oxygenated OA (25.0%), and biomass burning OA (13.8%). In contrast, Zheng et al. (2016)
have found that organic matter (OM) is the most abundant component (18 – 60%) in PM$_{2.5}$, and its relative contribution usually
decreases as the pollution level enhances whereas the contributions of secondary inorganic species (e.g, sulfate and nitrate) are
increased. Furthermore, air pollution in Beijing is strongly influenced by regional transport of pollutants and variations of
meteorological conditions (e.g., Li and Han, 2016; Bei et al., 2017). Zhong et al., (2018) have indicated that heavy pollution
episodes in Beijing can be generally divided into two phases of transport stage, which is characterized by the rising processes
mainly caused by pollutants transported from south of Beijing, and the cumulative stage, in which the cumulative explosive
growth of PM$_{2.5}$ is dominated by the stagnant meteorological conditions.



In recent years, the Chinese government usually takes some temporary control measures to ensure good air quality for some
important conferences or festivals held in Beijing, e.g., 2008 Summer Olympic Games, 2014 Asia-Pacific Economic
Cooperation (APEC) summit, and 2015 Victory Day parade (VDP). These actions provide valuable opportunity to evaluate
the effectiveness of emission controls on air pollution, which is certainly great value for future policy making. Numerous
studies have demonstrated that the temporary aggressive control measures are efficient in reducing primary pollutants and
secondary aerosol formation in Beijing (e.g., Wang et al., 2010; Guo et al., 2013; Li et al., 2015; Tao et al., 2016; Xu et al.,
2017). Furthermore, several studies have used models to quantify the contributions of pollution control strategies and
meteorological conditions in reducing pollutants, and they indicate that although the emission controls are effective in
decreasing aerosol concentrations, the meteorological conditions also play an important role in producing low aerosol loadings
(Gao et al., 2011; Liang et al., 2017). For example, Liang et al. (2017) have found that meteorological conditions and emission
control measures have comparable contributions in reducing $PM_{2.5}$ loadings in Beijing during the 2014 APEC (30 versus 28%,
respectively) and the 2015 VDP (38 versus 25%).
Autumn is a transition season between summer and winter and always has complex and variable meteorological conditions.
For example, Zhang et al. (2018) have reported that two typical weather patterns of the Siberian high and uniform high-pressure
field and the cold front and low-voltage system play important roles in causing heavy pollution episodes in Beijing in October.
Up to now, although plenty of studies have investigated the effectiveness of rigorous regulations on air pollutants in Beijing
during the Olympics, APEC, and VDP, there are still lack of investigations made in mid-autumn (e.g., October). In this study,
a series of measurements were performed in a regional background site in the Beijing-Tianjin-Hebei (BTH) region to
investigate the changes of $PM_{2.5}$ during the 19th National Congress of the Communist Party of China (NCCPC), which was
held in Beijing during 18 – 24 October. Some temporary control measures (e.g., restricting the number of vehicles, prohibiting
construction activities, and shutting down part of factories or restricting industrial production) were implemented in Beijing
and its vicinity. Consequently, the primary objectives of this study are (1) to investigate the effectiveness of emission control
measures on $PM_{2.5}$ and its chemical composition; (2) to determine the contributions of emission sources to $PM_{2.5}$ mass during
the NCCPC control and non-control periods; and (3) to evaluate the impacts of reductions of $PM_{2.5}$ on the aerosol direct
radiative forcing (DRF) at the Earth's surface. This study would further provide valuable information and dataset to help
controlling air pollution in Beijing.
**2 Materials and methods**
**2.1 Sampling site**
Intensive measurements were conducted from 12 October to 4 November 2017 at the Xianghe Atmospheric Observatory (39.75°
N, 116.96° E; 36 m a.s.l.) to investigate the characteristics of $PM_{2.5}$ and its radiative effects during the NCCPC period. Xianghe



is a small county with 0.33 million residents. It is located in a major plain-like area and is ~50 km southeast from Beijing and
~70 km north from Tianjin (Figure 1). This observation site is a regional aerosol background site that is influenced by mixed
emission sources from the BTH region. The sampling site is surrounded by residential areas and farmland and is ~5 km west
of Xianghe city center. More detailed description of the site may be found in Ran et al. (2016).

**2.2 Measurements**

**2.2.1 Offline measurements**

PM$_{2.5}$ samples were collected on 47 mm quartz-fiber filter (QM/A; Oregon, USA) and Teflon® filters (Whatman Limited,
Maidstone, UK) using two parallel mini-volume samplers (Airmetrics, Oregon, USA) at a flow rate of 5 L min$^{-1}$. The duration
of sampling was 24 h from 09:00 local time to 09:00 the next day. The quartz-fiber filters were used for analyses of water-
soluble inorganic ions and carbonaceous species, while the Teflon filters were used for inorganic elemental analysis. The PM$_{2.5}$
mass on each sample was weighed by a Sartorius MC5 electronic microbalance with ± 1 μg sensitivity (Sartorius, Göttingen,
Germany). Moreover, field blanks (a blank quartz-fiber filter and a blank Teflon filter) were collected and analysed to eliminate
the possible background effects.
Water-soluble inorganic ions, including F$^-$, Cl$^-$, NO$_3^-$, SO$_4^{2-}$, Na$^+$, K$^+$, Mg$^{2+}$, Ca$^{2+}$, and NH$_4^+$ were analyzed by a Dionex 600
ion chromatograph (IC, Dionex Inc., Sunnyvale, CA, USA). The four anions were separated using an ASII-HC column (Dionex
Corp.) and 20 mM potassium hydroxide as the eluent. The five cations were separated with a CS12A column (Dionex) and an
eluent of 20 mM methane sulfonic acid. More detailed description of the IC analyses may be found in Zhang et al. (2011).
Carbonaceous species, including organic carbon (OC) and elemental carbon (EC) were determined using a Desert Research
Institute (DRI) Model 2001 thermal/optical carbon analyzer (Atmoslytic Inc., Calabasa, CA, USA) following the Interagency
Monitoring of Protected Visual Environments (IMPROVE_A) protocol (Chow et al., 2007). Different concentration gradients
of standard sucrose solution were used to establish a standard carbon curve before analysis. Moreover, replicate analyses were
performed at a rate of one sample in every ten samples, and the repeatability is better than 15% for OC and 10% for EC in this
study. More information of the measurements may be found in Cao et al. (2003). Thirteen elements were determined by an
energy-dispersive X-ray fluorescence (ED-XRF) spectrometry (Epsilon 5 ED-XRF, PANalytical B.V., Netherlands), and these
elements included Al, Si, K, Ca, Ti, Cr, Mn, Fe, Cu, Zn, As, Br, and Pb. The analytical accuracy for ED-XRF measurements
was determined with a NIST Standard Reference Material 2783 (National Institute of Standards and Technology, Gaithersburg,
MD, USA). More detailed description of the ED-XRF may be found in Xu et al. (2012).

**2.2.2 Online measurements**

The aerosol optical properties were determined using a Photoacoustic Extinctiometer (PAX, Droplet Measurement
Technologies, Boulder, CO, USA) at a wavelength of 532 nm. The PAX measures light scattering (b$_{scat}$) and absorption (b$_{abs}$)



coefficients (in Mm⁻¹) simultaneously using a built-in wide-angle integrating reciprocal nephelometer and an aucoustic
technique, respectively. Before and during the sampling, $b_{scat}$ and $b_{abs}$ of the PAX were calibrated with a set of different
concentration gradients of ammonium sulfate and fullerene soot particles, respectively, which were generated with an atomizer
(Model 9302, TSI Inc., Shoreview, MN, USA). Detailed calibration procedure is described in Wang et al. (2018a; 2018b). In
this study, the inlet of PAX was installed with a $PM_{2.5}$ cutoff, and the sampled particles were dried by a Nafion® dryer (MD-
700-24S-1; Perma Pure, LLC., Lakewood, NJ, USA). The time resolution of data log was set to 1 minute.
One-minute average mixing ratios of NOx (NO + NO₂), O₃, and SO₂ were measured using a Model 42$i$ gas-phase
chemiluminescence NOx analyzer (Thermo Scientific, Inc., USA), a Model 49$i$ photometric ozone analyzer (Thermo Scientific,
Inc.), and a Model 43$i$ pulsed UV fluorescence analyzer (Thermo Scientific, Inc.), respectively. Standard reference NO, O₃,
and SO₂ gases were used to calibrate the NOx, O₃, and SO₂ analyzers, respectively, before and during the campaign. All the
online data were averaged to 24 h according to the duration of the filter sampling.
**2.2.3 Complementary data**
Wind speed (WS) and relative humidity (RH) were measured with the use of an automatic weather station installed at the
Xianghe Atmospheric Observatory. The weather charts at the surface for East Asia were accessible from the Korea
Meteorological Administration. The three-day backward trajectories and mixed layer height (MLH) were derived from the
Hybrid Single-Particle Lagrangian Integrated Trajectory (HYSPLIT) model (Draxler and Rolph, 2003), which is developed
by the National Oceanic and Atmospheric Administration (NOAA). The aerosol optical depth (AOD) was measured using a
sunphotometer (Cimel Electronique, Paris, France), and these data can be obtained freely from the Aerosol Robotic Network
data archive (http://aeronet.gsfc.nasa.gov). The fire counts were obtained from the Moderate Resolution Imaging
Spectroradiometer (MODIS) observations on Aqua and Terra satellites (https://firms.modaps.eosdis.nasa.gov/map).
**2.3 Data analysis methods**
**2.3.1 Chemical mass closure**
The chemically reconstructed $PM_{2.5}$ mass was calculated as the sum of OM, EC, $SO_4^{2-}$, $NO_3^-$, $NH_4^+$, $Cl^-$, fine soil, and trace
elements. Based on the results of Xu et al. (2015), a factor of 1.6 was adopted to convert OC to OM (OM = 1.6 × OC) to
account for those unmeasured atoms in organic materials. The mass concentration of fine soil was calculated by summing Al,
Si, K, Ca, Ti, Mn, and Fe oxides, and the equation is as follows (Cheung et al., 2011):
$[Fine\ soil] = [Al_2O_3] + [SiO_2] + [K_2O] + [CaO] + [TiO_2] + [MnO_2] + [Fe_2O_3] = 1.89 \times [Al] + 2.14 \times [Si] + 1.21 \times$
$[K] + 1.4 \times [Ca] + 1.67 \times [Ti] + 1.58 \times [Mn] + 1.43 \times [Fe]$ (1)





The mass concentration of trace elements was estimated as the sum of the elements that did not be used in the calculation of
fine soil:
[Trace elements] = [Cr] + [Cu] + [Zn] + [As] + [Br] + [Pb]                                              (2)
As shown in Figure S1, the reconstructed PM$_{2.5}$ mass is strongly correlated (r = 0.98) with the value from gravimetric
measurement, suggesting strong reliability of the chemical reconstruction method. The slope of 0.86 indicates that our
measured chemical species account for most of the PM$_{2.5}$ mass. The discrepancy between the reconstructed and measured
PM$_{2.5}$ mass was defined as "others".
**2.3.2 Receptor model source apportionment**
Positive matrix factorization (PMF) has been widely used in source apportionment studies in the past two decades (e.g., Cao
et al., 2012; Xiao et al., 2014; Tao et al., 2014; Huang et al., 2017). The principles of PMF are described elsewhere (Paatero
and Tapper, 2006). Briefly, PMF is a bilinear factor model that decomposes initial chemically speciated dataset into factor
contributions matrix $G_{ik}$ ($i \times k$ dimensions) and factor profiles matrix $F_{kj}$ ($k \times j$ dimensions), and then iteratively minimizes the
object function $Q$:
$X_{ij} = \sum_{k=1}^{p} G_{ik} F_{kj} + E_{ij}$                                                         (3)
$Q = \sum_{i=1}^{m} \sum_{j=1}^{n} (\frac{E_{ij}}{\sigma_{ij}})^2$                                         (4)
where $X_{ij}$ is the concentration of the $j$th species that measured in the $i$th sample; $E_{ij}$ is the model residuals; and $\sigma_{ij}$ represents
the uncertainty.
In this study, the PMF Model version 5.0 (PMF 5.0) from US Environmental Protection Agency (EPA) (Norris et al., 2014)
was employed to identify the source factors that contributed to PM$_{2.5}$ mass. Four to nine factors were extracted to determine
the optimal number of factors with random starting points. When the values of scaled residuals for all chemical species vary
between -3 and +3 and small $Q_{true}/Q_{expect}$ is obtained, the base run could be considered as stable. Further, the bootstrap analysis
(BS), displacement analysis (DISP), and bootstrap-displacement analysis (BS-DISP) were applied to assess the variability and
stability of the results. More detailed description of the determination methods of uncertainties for PMF solution can be found
in Norris et al. (2014).
**2.3.3 Regional chemical dynamical model**
The WRF-Chem (Weather Research and Forecasting model coupled to chemistry) is a 3-D online-coupled meteorology and
chemistry model, and it is used to simulate the formation processes of high PM$_{2.5}$ loadings in the BTH region during pollution
episodes after the NCCPC. WRF-Chem includes the components of meteorological processes (e.g., clouds, boundary layer,





temperature, and winds), pollutant emissions, chemical transformation, transport (e.g., advection, convective, and diffusive),
photolysis and radiation, dry and wet deposition, and aerosol interactions. Detailed description of WRF-Chem model may be
found in Li et al. (2011a; 2011b; 2012). Grid cells of 280 × 160 covering China with a horizontal resolution of 0.25° were
simulated. Twenty-eight vertical layers were set from the Earth's surface up to 50 hPa, and seven layers < 1 km were
established to ensure a high near-ground vertical resolution. The meteorological initial and boundary conditions were retrieved
from the National Centers for Environmental Prediction (NCEP) reanalysis dataset, and the chemical initial and boundary
conditions were obtained from the 6 h output of MOZART (Model for Ozone and Related chemical Tracers) (Emmons et al.,

87    2010).

In this study, the mean bias (MB), root mean square error (RMSE), and index of agreement (IOA) are used to evaluate the
performance of WRF-Chem simulation. The IOA is representative of the relative difference between the predicted and
measured values, and it varies from 0 to 1, with 1 indicating perfect performance of model prediction. These parameters were
calculated using the following equations (Li et al., 2011a):
$\text{MB} = \frac{1}{N}\sum_{i=1}^{N}(P_i - O_i)$         (5)
$\text{RMSE} = [\frac{1}{N}\sum_{i=1}^{N}(P_i - O_i)^2]^{\frac{1}{2}}$         (6)
$\text{IOA} = 1 - \frac{\sum_{i=1}^{N}(P_i - O_i)^2}{\sum_{i=1}^{N}(|P_i - P_{ave}| + |O_i - O_{ave}|)^2}$         (7)
where $P_i$ and $P_{ave}$ represent each predicted $PM_{2.5}$ mass concentration and their average values, respectively; $O_i$ and $O_{ave}$ are
each observed $PM_{2.5}$ mass concentration and their average values, respectively; and $N$ is representative of total number of the
predictions used for comparison.
**2.3.4 Calculation of chemical $b_{scat}$ and $b_{abs}$**
To determine the contributions of $PM_{2.5}$ to particles' optical properties, the $b_{scat}$ and $b_{abs}$ were reconstructed based on the major
chemical composition in $PM_{2.5}$ using the revised IMPROVE equations as follows (Pitchford et al., 2007):
$b_{scat} \approx 2.2 \times f_S(RH) \times [(NH_4)_2SO_4]_{Small} + 4.8 \times f_L(RH) \times [(NH_4)_2SO_4]_{Large} + 2.4 \times f_S(RH) \times [NH_4NO_3]_{Small} +$
$5.1 \times f_L(RH) \times [NH_4NO_3]_{Large} + 2.8 \times [OM]_{Small} + 6.1 \times [OM]_{Large} + 1 \times [Fine\ soil]$         (8)
$[X]_{Large} = \frac{[X]^2}{20\ \mu g\ m^{-3}}, for\ [X] < 20\ \mu g\ m^{-3}$         (9)
$[X]_{Large} = [X], for\ [X] \geq 20\ \mu g\ m^{-3}$         (10)
$[X]_{Small} = [X] - [X]_{Large}$         (11)





where the mass concentrations of ammonium sulfate ($[(NH_4)_2SO_4]$) and ammonium nitrate ($[NH_4NO_3]$) were estimated by
multiplying the concentrations of $SO_4^{2-}$ and $NO_3^-$ by a factor of 1.375 and 1.29, respectively (Tao et al., 2014); f(RH) is the
water growth for small and large mode of $(NH_4)_2SO_4$ and $NH_4NO_3$; and [X] represents the $PM_{2.5}$ composition involved in Eq.
(8). This analysis is based on the premise of particles being externally mixed. More detailed information of the IMPROVE
algorithms is described in Pitchford et al. (2007).
Considering that EC is the dominant species absorbing light in the visible region (e.g., 532 nm) (Massabò et al., 2015), the
relationship between $b_{abs}$ and EC mass concentration is determined by a linear regression:
$$b_{abs} = a \times [EC] + b \tag{12}$$
**2.3.5 DRF calculation**
The Tropospheric Ultraviolet and Visible (TUV) radiation model developed by the National Center for Atmospheric Research
(NCAR) is used to estimate the aerosol DRF for 180 – 730 nm at the Earth's surface. A detailed description of the model may
be found in Madronich (1993). Aerosol DRF is mainly affected by the aerosol column burden and its chemical composition,
which can be reflected by the parameters of AOD, aerosol absorption optical depth (AAOD), and single-scattering albedo
(SSA = (AOD-AAOD)/AOD). Based on the established relationship between AOD measured with the sunphotometer and
light extinction coefficient ($b_{ext} = b_{scat} + b_{abs}$) observed with the PAX, an effective height can be retrieved to convert the
IMPROVE-based chemical $b_{ext}$ to the AOD or AAOD that caused by the chemical species in $PM_{2.5}$. Due to the influences of
hygroscopic properties of $PM_{2.5}$, the measured dry $b_{ext}$ values here were modified to the wet $b_{ext}$ based on the water growth
function of particles described in Malm William et al. (2003). It should be noted that the estimated chemical AOD values are
based on the assumption that the aerosols are distributed homogeneously within a given effective height. Finally, the calculated
chemical AOD and SSA caused by different $PM_{2.5}$ composition were used in the TUV model to obtain the shortwave radiative
flux. The surface albedo, another influential factor for estimation of DRF, was obtained from the MOD43B3 product measured
with the Moderate Resolution Imaging Spectroradiometer (https://modis-atmos.gsfc.nasa.gov/ALBEDO/index.html). The
solar component in the TUV model was calculated in view of the δ-Eddington approximation, and the vertical profile of $b_{ext}$
used in the model is described in Palancar and Toselli (2004). The aerosol DRF is defined as the difference between the net
shortwave radiative flux with and without aerosol as follows:
$$DRF_{surface} = Flux\ (net)_{with\ aerosol,\ surface} - Flux\ (net)_{without\ aerosol,\ surface} \tag{13}$$





## 3 Results and discussion

### 3.1 Effectiveness of the control measures on reducing PM$_{2.5}$

Based on the dates of emission control measures, we divided the whole study period into two phases: the NCCPC control period from 12 to 24 October and non-control period from 25 October to 4 November. Temporal variations of mass concentrations of PM$_{2.5}$ and its major components during the two phases are shown in Figure 2, and a statistical summary of those data is presented in Table 1. The mass concentrations of PM$_{2.5}$ remain consistently low, generally < 75 μg m$^{-3}$ (NAAQS II) during the NCCPC control period, but the high loadings with PM$_{2.5}$ > 75 μg m$^{-3}$ are frequently observed during the non-control period. On average, the mass concentration of PM$_{2.5}$ during the NCCPC control period is 57.9 ± 9.8 μg m$^{-3}$, which is decreased by 31.2% compared with the non-control period (84.1 ± 38.8 μg m$^{-3}$). Compared with the previous important events that implemented pollution control measures in Beijing and its surrounding areas, the reduction ratio of PM$_{2.5}$ in the present study falls within the low limit reducing range of 30 – 50% for the Olympic Games (Wang et al., 2009; Li et al., 2013), but is lower than the range of 40 – 60% for the APEC period (Tang et al., 2015; Tao et al., 2016; J. Wang et al., 2017) and the range of 60 – 70% for the VDP period (Han et al., 2016; Liang et al., 2017; Lin et al., 2017).

As shown in Figure 2 (right panel), the chemical mass closure of PM$_{2.5}$ reveals that on average OM is the largest contributor (30.4%) to PM$_{2.5}$ mass during the non-control period, followed by NO$_3^-$ (16.7%), fine soil (11.2%), and EC (7.6%). In contrast, OM (24.3%) and NO$_3^-$ (22.9%) both dominate the PM$_{2.5}$ mass during the NCCPC control period, followed by SO$_4^{2-}$ (9.8%), NH$_4^+$ (9.1%), and EC (7.9%). The mass concentration of OM is decreased largely by 43.1% from 24.6 μg m$^{-3}$ during the non-control period to 14.0 μg m$^{-3}$ during the NCCPC control period. For the secondary water-soluble inorganic ions, the average mass concentrations of NO$_3^-$ (13.4 μg m$^{-3}$ versus 16.9 μg m$^{-3}$) and NH$_4^+$ (5.4 versus 6.8 μg m$^{-3}$) decrease by 20.7% and 20.6% during the NCCPC control period compared with the non-control period, respectively. However, SO$_4^{2-}$ exhibits similar level between the NCCPC control (5.8 μg m$^{-3}$) and the non-control (5.3 μg m$^{-3}$) periods. This may be attributed to the low SO$_2$ concentrations (10.3 ± 3.5 μg m$^{-3}$, Figure S2) during the entire campaign, which may not provide enough gaseous precursor to form substantial sulfate. Furthermore, the loadings of EC, Cl$^-$, and fine soil are reduced by 25.0, 44.8, and 40.8% during the NCCPC control period compared with the non-control period, respectively. The different reductions for each chemical species revealed their distinct responses to the emission controls and meteorological conditions.

As shown in Figure S2, both WSs (0.7 ± 0.3 versus 1.3 ± 0.8 m s$^{-1}$) and MLHs (304.3 ± 60.6 versus 373.7 ± 217.9 m) are lower for the NCCPC control period than the non-control period. This indicates that horizontal and vertical diffusion conditions during the NCCPC control period should be worse than the non-control period. Therefore, it is necessary to consider WS and MLH when further evaluates the effectiveness of pollution control measures. A simple and effective way is to compare the concentrations of air pollutants under stable atmospheric conditions (Wang et al., 2015; Liang et al., 2017). In this study, we defined the stable atmospheric conditions based on the correlations between PM$_{2.5}$ mass concentration and WS and MLH. As



163 shown in Figure 3, PM$_{2.5}$ mass concentration exhibits a power function relationship with WS (r = -0.65) and MLH (r = 0.77).

164 The criterion for judging stable conditions is whether the WS and MLH are lower than the values of turning points, which are

165 the slopes changed from large to relatively small values. However, there is no inflection point for power function, thus, we

166 used piecewise functions to determine the turning points. As shown in Figure 3, the intersections of two linear regressions can

167 be representative of turning points of the influences of meteorological conditions on PM$_{2.5}$ mass. Finally, the days with WS <

168 0.4 m s$^{-1}$ and MLH < 274 m are considered to have stable atmospheric conditions. There are two days for the NCCPC control

169 period and three days for the non-control period that satisfy the criterion. As shown in Table 1, the reduction ratios for PM$_{2.5}$

170 (43.4%) as well as NO$_3^-$ (25.9%), OM (68.1%), EC (40.0%), and fine soil (58.7%) are larger for only considering the days

171 with stable meteorological conditions compared with those for all days. The results further suggest that control measures have

172 great effectiveness in preventing pollution.

**173 3.2 Estimation of source contributions**

174 The mass concentrations of water-soluble inorganic ions (SO$_4^{2-}$, NO$_3^-$, NH$_4^+$, K$^+$, and Cl$^-$), carbonaceous (OC and EC), and

175 elements (Al, Si, Ca, Ti, Cr, Mn, Fe, Cu, Zn, As, Br, and Pb) were used as the PMF 5.0 model inputs. After compared the PMF

176 profiles with the reference profiles from previous literatures, the finally identified sources are (i) coal combustion, (ii) traffic-

177 related emissions, (iii) secondary inorganic aerosols, (iv) biomass burning, (v) industrial processes, and (vi) mineral dust. As

178 shown in Figure S3, the PMF modelled PM$_{2.5}$ mass concentrations are strongly correlated with the observed values (r = 0.98)

179 with a slope of 0.94, and simultaneously, the concentrations of each modelled chemical species represent goodness-of-fit of

180 linear regression with the measured values (r = 0.68 – 0.99) (Table S1). The results reveal that the six identified sources could

181 be reasonably physically interpretable profiles in this study.

182 Figure 4 presents the source profiles and the average contribution of each source to PM$_{2.5}$ mass during the NCCPC control and

183 non-control periods. As shown in Figure 4a, the first source factor is enriched in As (38.8%), Pb (32.9%), and Fe (30.3%) as

184 well as moderate contributions from Mn (26.2%), Zn (23.8%), Si (23.1%), and Ca (22.8%). The As has been proposed as a

185 useful tracer for coal burning (Hsu et al., 2009; Y. Chen et al., 2017). Moreover, the metals of Pb, Fe, Mn, and Zn could be

186 produced from the processes of coal combustion (Xu et al., 2012; Men et al., 2018), while Ca and Si can be consisted in coal

187 fly ash (Pipal et al., 2011). Thus, this source factor is assigned to the coal burning. There is no significant difference in PM$_{2.5}$

188 mass from coal burning between the NCCPC control (8.5 μg m$^{-3}$) and non-control (7.8 μg m$^{-3}$) periods. This may be due to

189 the fact that coal burning is mainly used for household energy for local residents, whereas the control measures do not involve

190 this sector. The PM$_{2.5}$ mass from this source is lower than the values (~20 – 60 μg m$^{-3}$) from coal burning in the BTH region

191 in winter (Huang et al., 2017), which can be explained by the increased domestic usage of coal for heating activities during

192 the cold season.


The second source factor is characterized by the elevated loadings of EC (42.1%) and Cu (40.7%) as well as moderate

contributions of OC (29.1%), Zn (27.1%), and Br (22.2%). Previous studies have indicated that carbonaceous aerosols are

strongly associated with gasoline and diesel exhaust (Cao et al., 2005), and thus, EC and OC can be used as indicators for

motor vehicle emissions (Chalbot et al., 2013; Khan et al., 2016a). Cu and Zn could be derived from accessories of vehicles,

such as lubricant oil, brake linings, metal brake wear, and tires (Lin et al., 2015). Br may be partly emitted from fuel combustion

in internal combustion engines (Bukowiecki et al., 2005). Therefore, the second source factor is representative of traffic-related

emissions. Furthermore, the mass concentration of $PM_{2.5}$ from this source is strongly correlated (r = 0.72) with the vehicle-

related NOx concentration (Figure S4), which further suggests the validity of the PMF-resolved source contributions. The
traffic-related emissions have similar percentages of contributions to $PM_{2.5}$ mass during the NCCPC control (14.8%) and non-
control (15.4%) periods (Figure 4c), but its mass concentration is 1.6 times lower for the NCCPC control period (8.9 μg m$^{-3}$)
than the non-control period (14.4 μg m$^{-3}$). This is attributed to the reduction of vehicle volume on road by traffic restriction
during the control period.
The third source factor is dominated by the high loadings of $SO_4^{2-}$ (45.4%), $NO_3^-$ (43.4%), and $NH_4^+$ (47.0%), and is obviously
classified as secondary inorganic aerosol (Zhang et al., 2013; Amil et al., 2016). Moreover, moderate loadings of As (30.5%),
Pb (27.4%), Cr (31.4%), Cu (30.7%), and EC (30.8%) are also assigned to this factor suggesting the influences of coal burning
and vehicle exhausts. Although the concentrations of gaseous precursors (e.g., $SO_2$ and NOx) during the NCCPC control period
are lower than the non-control period (Figure S2), the average mass contribution of $PM_{2.5}$ from source of secondary inorganic
aerosol is larger during the NCCPC control period (22.5 versus 18.3 μg m$^{-3}$), and this source becomes the largest contribution
factor (37.3% of $PM_{2.5}$ mass) during the NCCPC control period. Compared with the low RH (69%) during the non-control
period, the higher RH (84%) during the NCCPC control period could promote formation of the secondary inorganic aerosols
through aqueous reactions (Sun et al., 2014).
The fourth source factor is characterized by the high contributions of $K^+$ (59.5%) with moderate loadings of $Cl^-$ (33.3%), OC
(28.5%), $NO_3^-$ (37.1%), $SO_4^{2-}$ (21.1%), and $NH_4^+$ (39.6%). $K^+$ is a good tracer for biomass burning (Zhang et al., 2013; Wang
et al., 2016b), and $Cl^-$ and OC are also related to this source (Tao et al., 2014; Huang et al., 2017). Consequently, this factor is
assigned to the biomass burning. Previous studies have found that $SO_2$ and $NO_2$ can be converted into sulfate and nitrate
aerosols on the surface of pre-existing KCl particles during the regional transport of biomass-burning emissions (Du et al.,
2011). The abundant $NO_3^-$, $SO_4^{2-}$, and $NH_4^+$ associated with this factor may be indicative of aged biomass-burning particles.
As shown in Figure 4c, biomass burning contributes substantially large to $PM_{2.5}$ mass during the NCCPC control (21.6%) and
non-control periods (27.3%). This is because that Hebei is a large corn and wheat producing province, and the straws are
commonly used as biofuels for residential cooking and heating purposes or directly open field burned in the rural areas (J.
Chen et al., 2017). However, the mass concentrations of $PM_{2.5}$ from this source is lower during the NCCPC control period



(13.0 µg m⁻³) than the non-control period (25.7 µg m⁻³), which can be explained by the control policy for forbidding the open
space biomass-burning activities during the NCCPC period. Because the control measures do not involve the household use
of biofuels, and thus, the high contribution of biomass burning can be still measured during the NCCPC control period.
The fifth source factor have high loadings on Zn (41.3%), Br (38.0%), Pb (19.9%), As (19.2%), Cu (17.5%), and Mn (19.1%),
and is thought to be associated with industrial process emissions (Q. Q. Wang et al., 2017; Sammaritano et al., 2018). This
source contributes 3.6 µg m⁻³ to PM$_{2.5}$ mass during the NCCPC control period, which is lower than the non-control period
(16.2 µg m⁻³) by a factor of 4.5, and its percentage of contribution also increases from 6.0 to 17.2% accordingly. The results
reveal the effectiveness of restrictions on industrial activities during the NCCPC control period. Iron and steel factory is one
of the most important industries in BTH region, and the scale of iron and steel productions accounts for 28.8% of the total
amount of China in 2016 (NBS, 2017). The sintering process in the iron and steel industries can produce plenty of heavy metal
pollutants including Zn, Pb, and Mn (Duan and Tan, 2013). Hence, the iron and steel industries in the BTH region may be
possibly important source amongst industrial processes during the non-control period.
The predominant species in the sixth source factor are Al (55.9%), Si (55.7%), Ca (52.6%), and Ti (36.7%), which is obviously
classified as mineral dust (Zhang et al., 2013; Tao et al., 2014; Kuang et al., 2015). This factor contributes 3.8 µg m⁻³ (6.3%
of PM$_{2.5}$ mass) and 11.2 µg m⁻³ (12.3%) to PM$_{2.5}$ mass during the NCCPC control and non-control periods, respectively. The
possible sources for causing mineral dust may include (i) natural dust, which contains Al, Si, and Ti (Milando et al., 2016), (ii)
construction dust, which includes Ca (Liu et al., 2017), and (iii) road dust, which refers to the traffic-related species, such as
Cu, Zn, Br, and EC (Khan et al., 2016b; Zong et al., 2016). Here the mineral dust factor do not contain any notable contributions
from the traffic-related species. Thus, this factor may be mainly influenced by the natural dust and construction activities. As
shown in Figure S5, WS is positively correlated well (r = 0.75) with PM$_{2.5}$ mass that is contributed by mineral dust. To reduce
the impact of winds on crustal dust resuspension, we only compared the days with low winds (< 1 m s⁻¹), and only two sampling
days of 28 and 29 October were excluded. The result reveals that the mass concentration of PM$_{2.5}$ from mineral dust is reduced
by 67.0% during the NCCPC control period (3.8 µg m⁻³) compared with the non-control period (11.5 µg m⁻³), indicating that
the restrictions on construction activities during the NCCPC period are great effectiveness.
**3.3 Exploring the pollution episodes after the NCCPC control period**
As shown in Figure 2 (left panel), there are two pollution episodes (PE1: 25 – 27 October and PE2: 31 October – November
1) occurred subsequently after the NCCPC control period, and the average mass concentrations of PM$_{2.5}$ are 117.5 and 124.5
µg m⁻³ on PE1 and PE2, respectively. For PE1, the secondary inorganic aerosol is the dominant source, accounting for 54.6%
of PM$_{2.5}$ mass (Figure 5a), of which the formation of NO$_3^-$ is the most important due to its largest contribution to PM$_{2.5}$ mass
(26.8%) (Figure 5b). The mass concentration of NO$_3^-$ increases from less than 10 µg m⁻³ before PE1 to greater than 25 µg m⁻³
during PE1 (Figure 2). Further, we quantified the molar ratio of NO$_3^-$ to NO$_2$ (NOR = n-NO$_3^-$/(n-NO$_2$ + n-NO$_3^-$)), which can





be used to reflect nitrogen partitioning between the particle and gas phases (Zhang et al., 2011). As shown in Figure 6a, the
mass concentration of PM$_{2.5}$ enhances with the increased NOR (r = 0.65) during the entire campaign, suggesting nitrate
formation plays an important role in accumulation of high PM$_{2.5}$ loadings. The NORs ranges from 0.32 to 0.71 during the PE1,
which is significant higher (*t*-test, p < 0.01) than those before (0.23 − 0.29) or after (0.03 − 0.10) PE1, reflecting stronger nitrate
formation during the pollution period. Furthermore, NOR exhibits an exponential increase with RH (r = 0.80, Figure 6b),
indicating high RH is in favour of aqueous nitrate reaction. Therefore, the higher RHs (91 − 93%) during the PE1 promote the
aqueous nitrate production compared with the periods before (80 − 86%) or after pollution (33 − 57%).
Furthermore, OM is the second largest contributor during the PE1, accounting for 22.9% of PM$_{2.5}$ mass. The widely used EC-
tracer method (Lim and Turpin, 2002) was applied to estimate the primary and secondary OA (POA and SOA), and the lowest
10% percentile of the measured OC/EC ratio is used to identify the primary OC/EC ratio (Zheng et al., 2015). The estimated
mass concentrations of POA and SOA are 17.2 and 9.7 µg m$^{-3}$ during the PE1, accounting for 63.9 and 36.1% of OM mass,
respectively. Photochemical oxidation and aqueous reactions are two of the major mechanisms for SOA formation (Hallquist
et al., 2009). To evaluate the roles of these chemical reactions, the EC-scaled concentrations of SOA (SOA/EC) was used to
eliminate the impacts of different dilution/mixing conditions on SOA loadings (Zheng et al., 2015). As shown in Figure 6c,
the SOA/EC increases (r = 0.65) with the enhanced Ox (NO$_2$ + O$_3$), which is a proxy for atmospheric aging caused by
photochemical reactions (Canonaco et al., 2015), but it shows a weak correlation with RH (r = -0.32) (Figure 6d). The results
indicate that photochemical reaction rather than aqueous phase oxidation may be the major reaction mechanism for SOA
formation in this study. Thus, the low contribution of SOA during the PE1 may be due to the low photochemical activity under
the pollution condition.
In contrast, OM (31.8%) is the most abundant species in PM$_{2.5}$ during the PE2, followed by NO$_3^-$ (19.2%) (Figure 5b). The
mass concentration of K$^+$ increases substantially from 0.1 µg m$^{-3}$ before the PE2 to 1.7 µg m$^{-3}$ during the PE2, indicating a
strengthening influence of biomass-burning emissions. Indeed, the results of PMF show that biomass burning is the largest
source that accounted for 36.0% of PM$_{2.5}$ mass during the PE2 (Figure 5a). The 72-h backward trajectory analysis shows that
a lot of air masses during the PE2 originate from or pass through the fire counts which are located in the Inner Mongolia and
Shanxi (see Figure 7), indicating the impacts of transport of biomass-burning emissions. The estimated SOA contributes 47.7%
of OM mass, reflecting secondary formation of organics plays an important role in aggravating pollution. It should be noted
that the mass concentration of SOA during the PE2 is higher than PE1. Since the oxidizing conditions are similar for both
pollution episodes (e.g., Ox: PE1 = 78.0 µg m$^{-3}$ and PE2 = 86.7 µg m$^{-3}$) (Figure S2), the larger SOA during the PE2 may be
attributed to the aged SOA formed from the transport of biomass-burning emissions.
Previous studies have indicated that meteorological conditions have great effects on accumulation pollutants (Bei et al., 2016).
The weather charts in Figure 8 were used to analyze the synoptic systems. Further, the WRF-Chem model was applied to



simulate the formation processes of PM$_{2.5}$ during the two pollution episodes (Figure 9). As shown in Figure S6, the predicted
PM$_{2.5}$ and its major chemical composition exhibit roughly similar trends with the observed values. The calculated MB and
RMSE for PM$_{2.5}$ are -6.8 and 32.8 µg m$^{-3}$, and the IOA is estimated to be 0.75, indicating that the genesis of the two pollution
episodes is captured by the WRF-Chem model even though the average mass concentration of predicted PM$_{2.5}$ is lower than
the observed value. The most possible reasons may be due to the uncertainties caused by the complex of meteorological fields,
which determine the transport, diffusion, and removal of air pollutants in the atmosphere (Bei et al., 2012), and the discrepancy
of the emission inventory of PM$_{2.5}$ among different years.
As shown in Figure 8, before PE1, a weak cold high-pressure system in Siberia is moving toward south on 22 October, and
the BTH region is dominated by a cold high-pressure system, which is conducive to maintaining the pollutants at a low level.
From 24 to 25 October, the BTH region is controlled by a weak high-pressure system that followed by a low-pressure system
on the rear, and this leads to a convergence zone of southern air flow in the BTH region, which provides a unfavorable
meteorological condition accumulating pollutants gradually (Figure 9). As shown in Figure S2, NOx concentration increases
from 71.6 µg m$^{-3}$ on 22 October to 147.6 µg m$^{-3}$ on 25 October, which provides a high level of gaseous precursor for formation
of large loadings of nitrate subsequently. During 26 – 28 October (PE1), the cold air is piled up in the BTH region, and then
the cold high-pressure system is strengthened gradually. The weather of BTH region is dominated by cloudy, high RH, and
low surface WSs at this moment. Those unfavourable meteorological conditions further aggravates the accumulation of
pollutants in this area (Figure 9), and the WRF-Chem simulation shows that the BTH region contributes 73.6% of PM$_{2.5}$ mass
during the PE1. On 29 October, the cold high-pressure system moves toward south, and the north winds increase. Those
favourable meteorological conditions lead to a dilution of the atmospheric pollutants and as a result lower PM$_{2.5}$ loadings in
the BTH region (Figure 9).
During 31 October – 1 November (PE2), the BTH region is again dominated by a weak high-pressure system and a convergence
of northern air flow which is caused by the front of weak high-pressure and the rear of low-pressure. The local pollutants in
the BTH region are accumulated under those weather conditions, even though the loadings of PM$_{2.5}$ are influenced by the long-
range transport of biomass-burning emissions as we discussed above. The WRF-Chem simulation reveals that the BTH region
contributes 46.9% to PM$_{2.5}$ mass, which is comparable to that from other regions (53.1%). After 2 November, the northern
cold high-pressure system moves toward south, and the winds become strong resulting in improvement of air quality gradually.
**3.4 Impacts of PM$_{2.5}$ emission reduction on aerosol radiative effects**
The aerosol DRF refers to the change in the radiative energy balance due to the scattering and absorption of sunlight by aerosols.
As shown in Figure S7a, the reconstructed chemical b$_{scat}$ correlates strongly (r = 0.91) with the observed b$_{scat}$ values, with a
slope of 0.90, suggesting that the IMPROVE-based method provides a good estimation of the chemical b$_{scat}$. Moreover, based
on the good relationship between the measured b$_{abs}$ and EC concentrations (r = 0.82, slope = 10.8, Figure S7b), EC can be used



to calculate the chemical $b_{abs}$. Figure 10a shows the average contributions of each chemical component in $PM_{2.5}$ to the chemical
$b_{ext}$. On average, OM is the largest contributor (43.5%) to the chemical $b_{ext}$ during the non-control period, followed by $NH_4NO_3$
(32.4%), EC (14.3%), $(NH_4)_2SO_4$ (7.6%), and fine soil (2.2%). In contrast, during the NCCPC control period, $NH_4NO_3$ is the
largest contributor to the chemical $b_{ext}$, amounting to 36.7% of $b_{ext}$, and it is followed by OM (33.3%), EC (16.2%), $(NH_4)_2SO_4$
(11.9%), and fine soil (1.9%). Compared with previous Olympics and APEC studies, different contributions of $PM_{2.5}$
components to $b_{ext}$ are found. For example, Li et al. (2013) have reported that $(NH_4)_2SO_4$ (41%) has the largest contribution to
$b_{ext}$ during the Olympics, followed by $NH_4NO_3$ (23%), OM (17%), and EC (9%); Zhou et al. (2017) have found that OM (49%)
is the largest contributor to $b_{ext}$, followed by $NH_4NO_3$ (19%), $(NH_4)_2SO_4$ (13%), and EC (12%). These differences may be
attributed to the different reductions in $PM_{2.5}$ chemical species and the variable RH among studies which can influence the
hygroscopic properties of sulfate and nitrate.
As shown in Figure S8, the AOD measured with sunphotometer correlates well with $b_{ext}$ under ambient condition, with a slope
(effect height) of 708 m and r = 0.78. Based on the average effective height, the estimated chemical AOD (AOD = 708 × $b_{ext}$
× $10^{-6}$) and SSA contributed by each major component in $PM_{2.5}$ were putted into the TUV model to calculate their DRF at the
Earth's surface. The estimated average DRF varies from -33.2 to -3.4 W $m^{-2}$ with an average of -16.5 ± 6.7 W $m^{-2}$ during the
entire campaign, and the average value is similar to the previous study of -13.7 W $m^{-2}$ for photosynthetically active radiation
which is estimated by the Santa Barbara DISORT Atmospheric Radiative Transfer model (SBDART) in Xianghe in autumn
(Xia et al., 2007a). Compared with previous DRF studies in the ultraviolet and visible region in China, the average DRF value
in this study is similar to the rural site of Taihu (-17.8 W $m^{-2}$, Xia et al., 2007b) but is less negative than the suburban or urban
sites of Linan (-73.5 W $m^{-2}$, Xu et al., 2003), Nanjing (-39.4 W $m^{-2}$, Zhuang et al., 2014), and Xi'an (-100.5 W $m^{-2}$, Wang et
al., 2016b), where high aerosol loadings are observed during their sampling periods.
As shown in Figure 10b, the estimated average DRF during the NCCPC control period is -14.0 ± 3.0 W $m^{-2}$, which is less
negative than the value during the non-control period (-19.3 ± 8.6 W $m^{-2}$). This should be attributed to the lower $PM_{2.5}$ loadings
during the NCCPC control period. The DRF reduction ratio (26.3%) during the NCCPC control period is smaller than the
value during the APEC period (61.3%, Zhou et al., 2017). Furthermore, the DRF values can be as high as -24.7 and -28.2 W
$m^{-2}$ during the PE1 and PE2, respectively. Figure 10b also shows the DRF caused by different types of chemical components
in $PM_{2.5}$. EC has the largest (most negative) effects on DRF at the surface during the non-control period, that is, a DRF value
of -13.4 W $m^{-2}$, followed by OM (-3.0 W $m^{-2}$), $NH_4NO_3$ (-2.2 W $m^{-2}$), $(NH_4)_2SO_4$ (-0.5 W $m^{-2}$), and fine soil (-0.15 W $m^{-2}$).
Due to the reduction of aerosol loadings during the NCCPC control period, the DRF values caused by EC, $NH_4NO_3$, OM, and
fine soil are decreased to -10.1, -1.7, -1.6, and -0.09 W $m^{-2}$, respectively, with corresponding reduced proportions of 24.6, 22.7,
46.7, and 40.0%. The results suggest that the short-term mitigation measures during the NCCPC control period would useful
for alleviating the cooling effects of $PM_{2.5}$ at the surface in Beijing.



## 4 Conclusions

In this study, we present an investigation of the impacts of short-term emission controls on the changes of PM$_{2.5}$ chemical composition and aerosol radiative effects at the Earth's surface during the 19th NCCPC period. The average mass concentration of PM$_{2.5}$ during the NCCPC control period is 57.9 ± 9.8 µg m$^{-3}$, which is decreased by 31.2% compared with the non-control period (84.1 ± 38.8 µg m$^{-3}$). The major chemical species of OM, NO$_3^-$, NH$_4^+$, EC, and fine soil are decreased by 43.1, 20.7, 20.6, 25.0, and 40.8% during the NCCPC control period compared with the non-control period, respectively. When considering the days with stable meteorological conditions, the reduction ratios of PM$_{2.5}$ (43.4%), NO$_3^-$ (25.9%), OM (68.1%), EC (40.0%), and fine soil (58.7%) are larger compared with those for all days. The results indicate that control measures have great effectiveness in preventing pollution. Further, the PMF receptor model shows that the biomass burning (27.3%) is the largest contributor to PM$_{2.5}$ mass during the non-control period, followed by secondary inorganic aerosol (19.5%), industry processes (17.2%), traffic-related emissions (15.4%), mineral dust (12.3%), and coal burning (8.3%). In contrast, secondary inorganic aerosol (37.3%) is the largest contributor to PM$_{2.5}$ mass, followed by biomass burning (21.6%), traffic-related emissions (14.8%), coal burning (14.1%), mineral dust (6.3%), and industry processes (6.0%). The mass concentrations of PM$_{2.5}$ contributed by traffic-related emissions, biomass burning, industry processes, and mineral dust are decreased during the NCCPC control period compared with the non-control period. However, there is no significant difference in PM$_{2.5}$ mass from coal burning between these two periods, while increased PM$_{2.5}$ mass concentration is found during the NCCPC control period for secondary inorganic aerosol.

There are two pollution episodes (PE1: 25 – 27 October and PE2: 31 October – November 1) occurred subsequently after the NCCPC control period with the average mass concentrations of 117.5 and 124.5 µg m$^{-3}$ on PE1 and PE2, respectively. For the PE1, the secondary inorganic aerosol is the dominant source, accounting for 54.6% of PM$_{2.5}$ mass, of which the formation of NO$_3^-$ is the most important due to its largest contribution to PM$_{2.5}$ mass (26.8%). High RH for PE1 is in favour of aqueous nitrate reaction. In contrast, OM (31.8%) is the most abundant species in PM$_{2.5}$ during the PE2, and the PMF shows that biomass burning is the largest source that accounts for 36.0% of PM$_{2.5}$ mass. The WRF-Chem simulation shows that the BTH region contributes 73.6% and 46.9% of PM$_{2.5}$ mass during the PE1 and PE2, respectively.

Based on the IMPROVE formula, OM is the largest contributor (43.5%) to the chemical b$_{ext}$ during the non-control period, followed by NH$_4$NO$_3$ (32.4%), EC (14.3%), (NH$_4$)$_2$SO$_4$ (7.6%), and fine soil (2.2%). During the NCCPC control period, NH$_4$NO$_3$ is the largest contributor amounting to 36.7% of b$_{ext}$, and it is followed by OM (33.3%), EC (16.2%), (NH$_4$)$_2$SO$_4$ (11.9%), and fine soil (1.9%). The TUV model shows that the estimated average DRF (-14.0 ± 3.0 W m$^{-2}$) at the surface during the NCCPC control period is 27.5% less negative than the non-control period (-19.3 ± 8.6 W m$^{-2}$), which is attributed to the lower PM$_{2.5}$ loadings during the NCCPC control period. Furthermore, EC has the largest (most negative) effects on DRF at the surface during the non-control period, that is, a DRF value of -13.4 W m$^{-2}$, followed by OM (-3.0 W m$^{-2}$), NH$_4$NO$_3$ (-2.2





W m$^{-2}$), (NH$_4$)$_2$SO$_4$ (-0.5 W m$^{-2}$), and fine soil (-0.15 W m$^{-2}$). Due to the reduction of aerosol loadings during the NCCPC
control period, the DRF values caused by EC, NH$_4$NO$_3$, OM, and fine soil are decreased to -10.1, -1.7, -1.6, and -0.09 W m$^{-2}$,
respectively, with corresponding reduced proportions of 24.6, 22.7, 46.7, and 40.0%. The results suggest that the short-term
mitigation measures during the NCCPC control period would useful for alleviating the cooling effects of PM$_{2.5}$ at the surface
in Beijing.
**Acknowledgments**
This work was supported by the National Research Program for Key Issues in Air Pollution Control and the National Natural
Science Foundation of China (41503118 and 41661144020). The authors are grateful to the staff from Xianghe Atmospheric
Observatory for their assistance with field sampling.

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



'50    **Table 1** Summary of PM$_{2.5}$ and its major chemical components during the NCCPC control and non-control periods in Xianghe.

| Components | Total average (µg m$^{-3}$) | Control period (µg m$^{-3}$) | Noncontrol period (µg m$^{-3}$) | Change ratio[a] (%) |
|---|---|---|---|---|
| PM$_{2.5}$ | 70.0 | 57.9 (63.7)[b] | 84.1 (112.6) | 31.2 (43.4) |
| NO$_3^-$ | 15.0 | 13.4 (18.0) | 16.9 (24.3) | 20.7 (25.9) |
| SO$_4^{2-}$ | 5.6 | 5.8 (7.6) | 5.3 (6.6) | -9.4 (-15.2) |
| NH$_4^+$ | 6.0 | 5.4 (8.6) | 6.8 (9.7) | 20.6 (11.3) |
| Cl$^-$ | 2.2 | 1.6 (1.5) | 2.9 (3.4) | 44.8 (55.9) |
| Organic matter | 18.9 | 14.0 (9.5) | 24.6 (29.8) | 43.1 (68.1) |
| Elemental carbon | 5.2 | 4.5 (4.5) | 6.0 (7.5) | 25.0 (40.0) |
| Trace elements | 1.8 | 1.4 (1.2) | 2.3 (3.0) | 39.1 (60.0) |
| Fine soil | 5.5 | 4.2 (2.6) | 7.1 (6.3) | 40.8 (58.7) |

'51    [a]([Non-control period]-[NCCPC control period])/[Non-control period].

'52    [b]Value in brackets represents the result from only considering the days with stable meteorological conditions (wind speed <

'53    0.4 m s$^{-1}$ and mixed layer height < 274 m).

'54



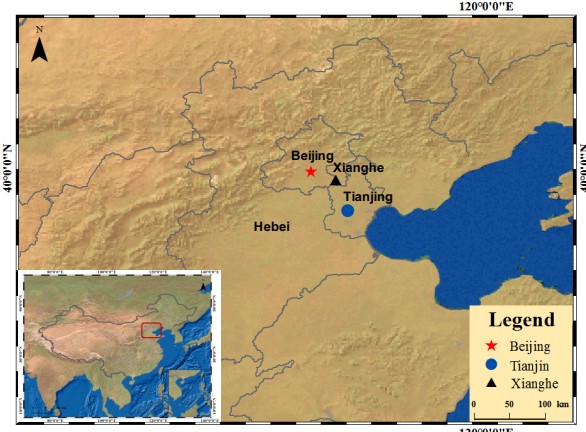

'55

'56

'57    **Figure 1.** Location of the Xianghe sampling site and surrounding areas. The map in the figure was drawn using the ArcGIS.

'58





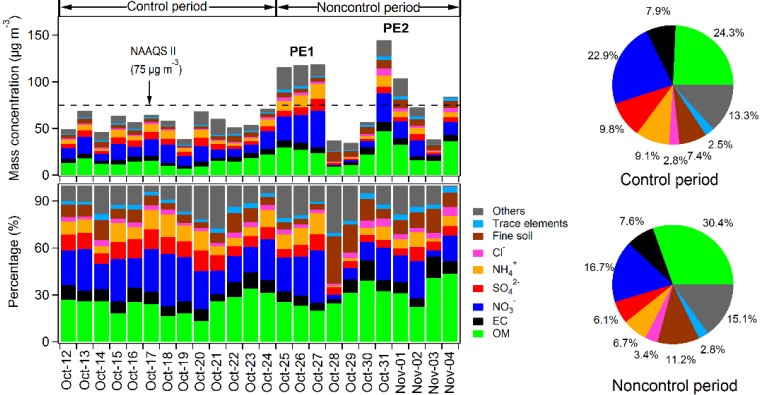

'59

'60

'61 **Figure 2.** (left panel) Daily variations of the contributions of chemical species to PM$_{2.5}$ mass during the entire campaign and

'62 (right panel) their average contributions during the NCCPC control and non-control period. PE1 and PE2 represent two

'63 pollution episodes.

'64





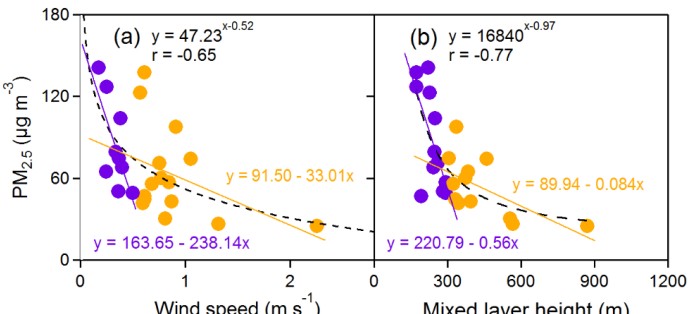

'65

'66

'67     **Figure 3.** Scatter plots showing the correlations between PM2.5 mass concentration and (a) wind speed and (b) mixed layer

'68     height. The purple and yellow scattered points represent different distribution areas.

'69





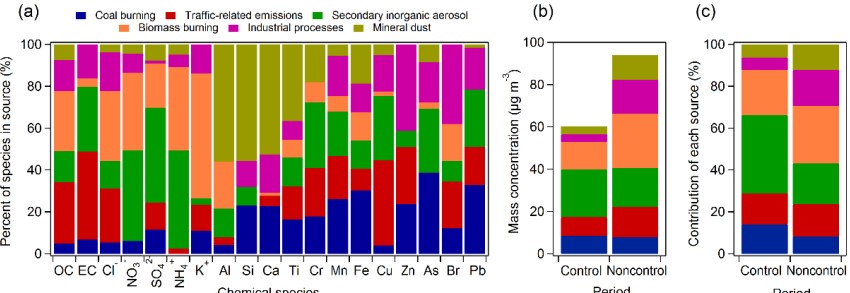

'70

'71

'72    **Figure 4.** (a) Source profiles for the six sources determined by the positive matrix factorization model version 5.0, (b) the

'73    mass concentrations of $PM_{2.5}$ contributed by each source, and (c) the average source contribution of each source to the $PM_{2.5}$

'74    mass.

'75





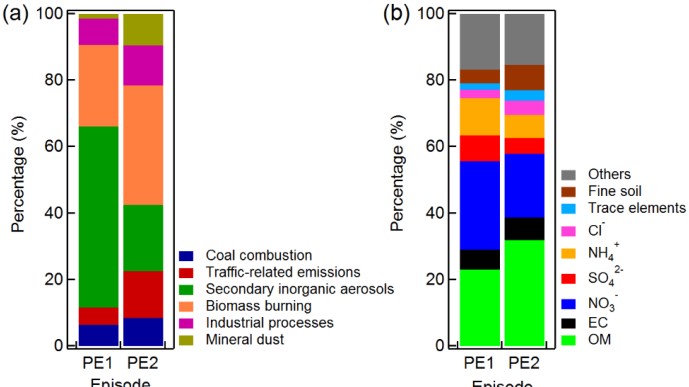

'76

'77

'78    **Figure 5.** Average source contributions of (a) each PMF source factor (see Figure 4) and (b) chemical species to the $PM_{2.5}$

'79    mass during two pollution episodes (PE1 and PE2).

'80





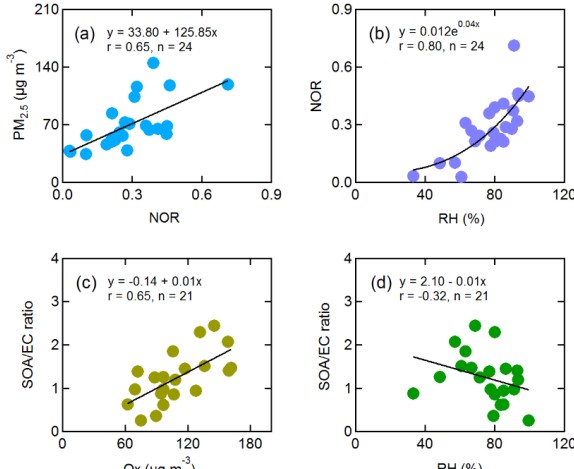

'81

'82

'83  **Figure 6.** Relationships between (a) $PM_{2.5}$ mass concentration and molar ratio of $NO_3^-$ to $NO_2$ (NOR), (b) NOR and relative

'84  humidity (RH), (c) the ratio of secondary organic aerosol and elemental carbon (SOC/EC) and Ox, and (d) SOA/EC and RH

'85  during the entire campaign.

'86





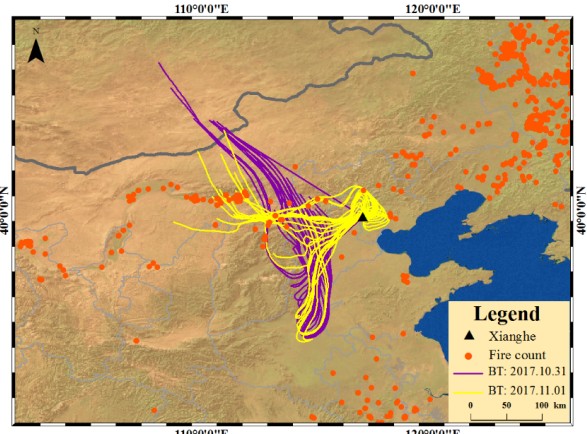

'87

'88

'89  **Figure 7.** Three-day backward air mass trajectories reaching at 150 m above ground every hour during 31 October to 1

'90  November 2017. The orange points represent the fire counts which derived from the Moderate Resolution Imaging

'91  Spectroradiometer observations.

'92




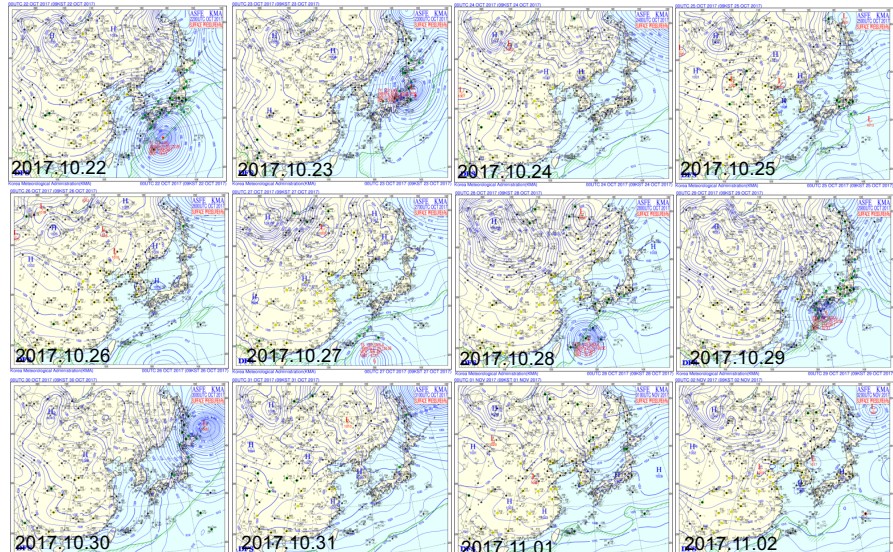

'93

'94

'95 **Figure 8.** Surface weather patterns at 08:00 (local time) over East Asia from 22 October to 2 November 2017.

'96





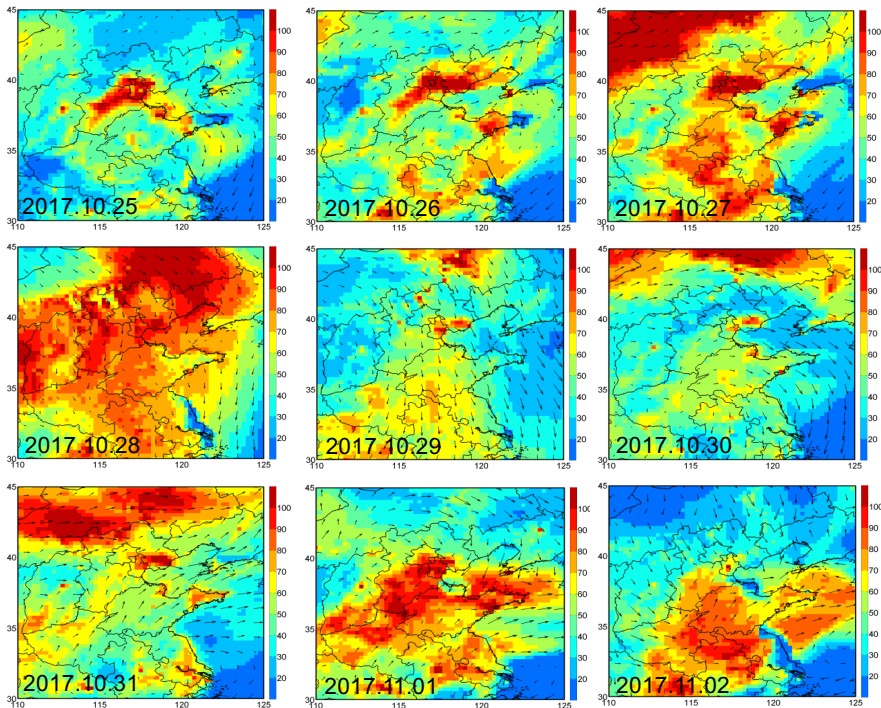

**Figure 9.** WRF-Chem simulated daily averaged PM$_{2.5}$ concentration (μg m$^{-3}$) in the Beijing-Tianjin-Hebei and its surrounding areas from 25 October to 2 November 2017.





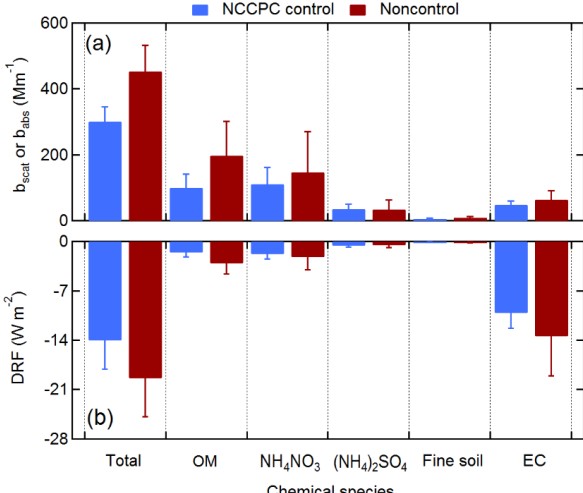

**Figure 10.** Average values of (a) light scattering ($b_{asct}$) or absorption ($b_{abs}$) coefficient and (b) direct radiative forcing (DRF)

at the surface contributed by each $PM_{2.5}$ chemical composition during the NCCPC control and non-control periods.