# Peer review of "Impacts of short-term mitigation measures on PM2.5 and radiative effects: a case study from a regional background site near Beijing, China"

_Atmospheric Chemistry and Physics, 2018_

## Referee Comment (RC1) · Chen (Referee) · 9 Dec 2018

This study was conducted in a regional background site near Beijing during the 19th National Congress of the Communist Party of China. The authors investigated the effectiveness of short-term mitigation measures on PM2.5 and aerosol direct radiative forcing. They found that PM2.5 mass and its sources are reduced significantly during the control period compared with the non-control period. Those decreases in aerosol concentration in turn, as shown by the climate radiative effect estimates, alleviated aerosol cooling effects. Moreover, the authors further analyzed two pollution episodes after control period based on the WRF-Chem model. This is an interesting

study. I believe that this paper makes a useful contribution to the literature and could be published in ACP after a minor revision in response to the following suggestions (see specific comments below). Specific Comments: 1. In section 2.2.1: the authors should give the storage condition of the samples. 2. Page 8, Line 11-13: It should be noted that the equation (12) is based on the assumption of no contribution from brown carbon, a light-absorbing organic matter. It should be pointed out this in the article. 3. Section 3.1: This study analyzed data from a single site near Beijing, even though it included detail chemical and optical measurements. The emission control for NCCPC control period included a wide range of measures and could impact the air quality for a lager domain. Therefore, it would be more convincing if the authors could also include measurements for surrounding areas from other platforms, such as the AERONET AODs and satellite aerosol retrievals. 4. About the light scattering construction of the particles (Sec 3.4, "Impacts of PM2.5 emission reduction on aerosol radiative effects"), the reconstructed bscat shows some deviation from the estimated bscat values. What is the reason for the difference? 5. The paper must be polished and edited for English grammar and word usage before it can be published in ACP.

---

## Referee Comment (RC2) · Anonymous Referee #2 · 27 Dec 2018

General comments

This manuscript attempts to examine the impacts of emission reduction on PM2.5 and radiative effects (surface DRF as the authors defined) using field measurements and WRF-Chem simulations at a regional background station in the Beijing-Tianjin-Hebei region in China. The impacts are examined by comparing the changes in observation-derived speciated PM2.5 concentrations and DRF during and after an emission-controlled period. The paper is reasonably written and results are reasonably presented, and it can be accept for publishing with revisions that address the following issues.

[Figure]

A major weakness in the study is that, as the main objective is to investigate the impacts of emission reduction measure on PM2.5 and DRF (emphasized in the title and abstract), this paper has a major flaw in separating the effects of emission reduction and meteorological conditions. Although the authors make an effort to make comparisons between the during- and-post-control periods under stable meteorological conditions, the determination of the "stable" conditions is quite rough, and it is not clear how similar the meteorological conditions are for the days selected for the comparison (even under stable conditions, the degree of the stability would significantly affect air quality). To separate these two factors, I would suggest the authors to do a more thorough analysis of the meteorological conditions, or ideally, based on the information they have and/or can obtain, construct an emission reduction scenario for the NCCPC control period and conduct additional WRF-Chem simulations and analysis.

Another issue is about the source apportionment in Section 3.2 using PMF. The authors assign the third source factor to secondary inorganic aerosols (SIA). This is not appropriate, since SIA is not an emission source, and it may have contributions from other sources they identify, such as coal combustion, mobile, industry, and biomass burning, i.e., SIA is not independent to other four identified anthropogenic emissions sources.

Specific comments

1. Page 8, line 13. It is better to show the regression results, and specify the values of a and b used.

2. Page 9, lines 52-54. Small changes in sulfates may also be attributed to small changes in SO2 emissions during the campaign.

3. Figures 4 and 5. Copy the source legend from Fig 5 to Fig 4.

4. Page 10, lines 64 -72. First, as pointed earlier, the approach to determine the "stable conditions" is rough. Second, the samples (3 days and 2 days) for the stable conditions

are too small, which would make the comparison statistically no meaningful. A better analysis is needed to separate the impacts of emission reduction and meteorological conditions.

5. Page 15, lines 42-43. It is surprising that with an averaged surface concentration of 6.0ug/m3, EC imposes the largest cooling effects in surface DRF during the non-control period and several factors higher than that of OM, while the light extinction by OM is much higher than by EC. An explanation would be helpful.

6. Fig 8 seems too small and a little bit complicated, which make it difficult to the reader to understand the effects of meteorological conditions on air quality in the BTH area. In addition, the location of the Xianghe site should be specified in the figure. Similar figure for October 12-23 might also be needed when you do analysis in decomposing the influences of the emission reduction and meteorology (especially for the five "stable" days).

Technical

The language need to be polished. The authors need go through the manuscript carefully and make edits. Following are just a few pickups.

Page 2 line 35, page 15 line 46, page 17 line 82: change "would" to "should"?

Page 2 line 44, change "experienced" to "experiencing"?

Page 3 line 78, change "low-voltage" to "low-pressure".

Page 14 line 88, "genesis"?

---

## Author Comment (AC1) · 23 Jan 2019

*This study was conducted in a regional background site near Beijing during the 19th National Congress of the Communist Party of China. The authors investigated the effectiveness of short-term mitigation measures on PM$_{2.5}$ and aerosol direct radiative forcing. They found that PM$_{2.5}$ mass and its sources are reduced significantly during the control period compared with the non-control period. Those decreases in aerosol concentration in turn, as shown by the climate radiative effect estimates, alleviated aerosol cooling effects. Moreover, the authors further analyzed two pollution episodes after control period based on the WRF-Chem model. This is an interesting study. I believe that this paper makes a useful contribution to the literature and could be published in ACP after a minor revision in response to the following suggestions (see specific comments below).*

**Response:** The authors appreciate the reviewer for his or her valuable time to review our manuscript.

*Specific Comments:*

*1. In section 2.2.1: the authors should give the storage condition of the samples.*

**Response:** We have added information concerning storage conditions for the samples in the revised manuscript. It now reads: "To minimize the evaporation of volatile materials, the samples were stored in a refrigerator at -4 °C before the chemical analyses."

*2. Page 8, Line 11-13: It should be noted that the equation (12) is based on the assumption of no contribution from brown carbon, a light-absorbing organic matter. It should be pointed out this in the article.*

**Response:** Following the reviewer's suggestion, we revised the text to read: "A second assumption for this part of the study was that there was negligible absorption by brown carbon in the visible region (Yang et al., 2009), and on this basis, the b$_{abs}$ can be determined from the EC mass concentration using linear regression (Eq. 12)."

*3. Section 3.1: This study analyzed data from a single site near Beijing, even though it included detail chemical and optical measurements. The emission control for NCCPC control period included a wide range of measures and could impact the air quality for a lager domain. Therefore, it would be more convincing if the authors could also include measurements for surrounding areas from other platforms, such as the AERONET AODs and satellite aerosol retrievals.*

**Response:** Although the AERONET AODs are helpful for providing a spatial distribution of aerosols in Beijing-Tianjin-Hebei (BTH) region, the observation periods were limited, typically at ~10:30 and 13:30 local time. Another complicating factor is

that relative humidity can have an important influence on AODs. After careful consideration, we concluded that it is more useful to focus on PM$_{2.5}$ concentrations at different locations in BTH region to illustrate the effectiveness of the control measures. As shown in Figure S3 (revised supporting information, also see Figure R1 below), the PM$_{2.5}$ concentrations over much of the BTH region showed a decreasing trend during the NCCPC-control period compared with the non-control period. In the revised manuscript, we added the following text: "Meanwhile, the PM2.5 mass concentrations obtained from the China Environmental Monitoring Center also showed a decreasing trend over most of the BTH region during the NCCPC-control period (see Figure S3)."

[Figure]

**Figure R1.** Spatial distribution of PM$_{2.5}$ mass concentration in Beijing-Tianjin-Hebei region during the (a) 19th National Congress of the Communist Party of China (NCCPC) control period and (b) non-control period.

*4. About the light scattering construction of the particles (Sec 3.4, "Impacts of PM$_{2.5}$ emission reduction on aerosol radiative effects"), the reconstructed b$_{scat}$ shows some deviation from the estimated b$_{scat}$ values. What is the reason for the difference?*

**Response:** Although the IMPROVE-based method provides reasonable estimates of the chemical b$_{scat}$ in this study, the lack of locally-derived mass scattering efficiency information is a probable reason for the ~10% underestimates of measured values. In the revised manuscript, we revised the text, which now reads: "This result indicates that the IMPROVE-based method provided a good estimation of the chemical b$_{scat}$; nonetheless, it is likely that more locally-measured mass scattering efficiencies for each chemical species could reduce the underestimates of measured values."

*5. The paper must be polished and edited for English grammar and word usage before it can be published in ACP.*

**Response:** Our revised manuscript has been polished by a native English speaker.

Please see our new manuscript.

---

## Author Comment (AC2) · 23 Jan 2019

*General comments*

*This manuscript attempts to examine the impacts of emission reduction on PM$_{2.5}$ and radiative effects (surface DRF as the authors defined) using field measurements and WRF-Chem simulations at a regional background station in the Beijing-Tianjin- Hebei region in China. The impacts are examined by comparing the changes in observation-derived speciated PM$_{2.5}$ concentrations and DRF during and after an emission-controlled period. The paper is reasonably written and results are reasonably presented, and it can be accept for publishing with revisions that address the following issues.*

**Response:** The authors appreciate the reviewer's thoughtful and valuable suggestions, and we believe that the revised manuscript has been significantly improved after considering his or her comments. Below are point-to-point responses.

*A major weakness in the study is that, as the main objective is to investigate the impacts of emission reduction measure on PM$_{2.5}$ and DRF (emphasized in the title and abstract), this paper has a major flaw in separating the effects of emission reduction and meteorological conditions. Although the authors make an effort to make comparisons between the during- and-post-control periods under stable meteorological conditions, the determination of the "stable" conditions is quite rough, and it is not clear how similar the meteorological conditions are for the days selected for the comparison (even under stable conditions, the degree of the stability would significantly affect air quality). To separate these two factors, I would suggest the authors to do a more thorough analysis of the meteorological conditions, or ideally, based on the information they have and/or can obtain, construct an emission reduction scenario for the NCCPC control period and conduct additional WRF-Chem simulations and analysis.*

**Response:** The reviewer correctly points out that variations in the mass concentrations of PM$_{2.5}$ and its chemical composition can be caused by a variety of factors, including meteorological conditions as well as emission sources. We agree with the reviewer that it would be desirable to construct an emission reduction scenario for the NCCPC-control period and then perform additional WRF-Chem simulations and analyses. Unfortunately, it was not possible for us to obtain detailed information concerning the reduction measures taken by the government, and therefore we could not develop an accurate emission inventory for the NCCPC-control period.

As an alternative, we compared days during the control and non-control periods with stable atmospheric conditions because that was a way to evaluate particle accumulation when the effects of transport would be minimal. Furthermore, because the duration of the control period was not long, it was not possible to precisely match meteorological conditions to investigate reduction in PM$_{2.5}$ during NCCPC-control and non-control period. Although "stable conditions" were empirically defined for our study, the general idea of minimizing meteorological influences was helpful for evaluating the

effectiveness of the emission control measures. We focused on wind speed and mixed layer height because they are important factors in determining the horizontal and vertical dispersion of particles.

As shown in Figure 3 (also see Figure R1 below) in the revised manuscript, the relationships between $PM_{2.5}$ concentrations and wind speed and mixed layer height can be fitted with power functions. Our strategy was to use the inflection points of the power functions as a way to identify stable atmospheric conditions. The average wind speeds and mixed layer heights were lower under stable atmospheric conditions during the NCCPC-control period than the non-control period, indicating that particles may have been more prone to accumulate during the NCCPC-control period. This means that if there had been no effective control measures during the NCCPC-period, the mass concentrations of $PM_{2.5}$ likely would have been higher compared with the days under stable atmospheric condition during the non-control period, but this was not the case. Thus, we think that the "stable atmospheric condition" approach is still useful for evaluating the effectiveness of the control measures.

Moreover, we now include surface weather charts in revised Figure S5 (also see Figure R2 below) to compare and contrast the weather conditions during the days with stable atmospheric conditions during the control and non-control periods. Finally, following the reviewer's suggestion, we include a more in-depth analysis of the meteorological conditions in the revised manuscript. The text now reads: "There were two days for the NCCPC-control period and three days for the non-control period that satisfied the stability criteria. The surface charts (Figure S5) show that the weather conditions for those selected stable atmosphere days during the NCCPC-control and non-control periods were mainly controlled by uniform pressure fields and weak low-pressure systems, respectively, and those conditions led to weak or calm surface winds. Due to the lower WS (0.2 versus 0.3 m s$^{-1}$) and MLH (213 versus 244 m) during the NCCPC-control period relative to the non-control period, the horizontal and vertical dispersion for the stable atmospheric days were slightly weaker during the NCCPC-control period. As shown in Table 1, the percent differences for $PM_{2.5}$ (43.4%), NO3- (25.9%), OM (68.1%), EC (40.0%), and fine soil (58.7%) were larger for the days with stable atmospheric conditions compared with those for all days. These results are a further indication that the control measures were effective in reducing pollution, but meteorology also influenced the aerosol pollution."

[Figure]

**Figure R1.** Scatter plots showing the relationships between PM$_{2.5}$ mass concentrations and (a) wind speed and (b) mixed layer height.

[Figure]

**Figure R2.** Surface weather charts for 08:00 (local time) over East Asia during the five

days with stable atmospheric conditions. The black triangles represent Xianghe.

*Another issue is about the source apportionment in Section 3.2 using PMF. The authors assign the third source factor to secondary inorganic aerosols (SIA). This is not appropriate, since SIA is not an emission source, and it may have contributions from other sources they identify, such as coal combustion, mobile, industry, and biomass burning, i.e., SIA is not independent to other four identified anthropogenic emissions sources.*

**Response:** In the broadest terms, $PM_{2.5}$ originates from primary sources (e.g., coal combustion, traffic emissions, industry, and biomass burning) and secondary processes, that is, the formation of particles through homogeneous reactions in the atmosphere. As the reviewer correctly noted, secondary inorganic aerosol forms from precursors emitted by primary sources. Receptor models (e.g., PMF) generally cannot resolve the sources for secondary particles, and therefore, we now classify this factor as "secondary particle formation" in the revised manuscript.

*Specific comments*
*1. Page 8, line 13. It is better to show the regression results, and specify the values of a and b used.*

**Response:** Following the reviewer's suggestion, we added the following in the revised manuscript: "As shown in Figure S2, the derived slope (a) and intercept (b) for the regression model were 10.8 $m^2$ $g^{-1}$ and -4.7, respectively."

*2. Page 9, lines 52-54. Small changes in sulfates may also be attributed to small changes in $SO_2$ emissions during the campaign.*

**Response:** Yes, in addition to the low $SO_2$ concentrations throughout the campaign, the change in $SO_2$ concentration during the NCCPC-control (8.5 µg $m^{-3}$) versus non-control period (12.4 µg $m^{-3}$) was small. Following the reviewer's suggestion, we revised the original explanation to "However, $SO_4^{2-}$ exhibited similar loadings during the NCCPC-control (5.8 µg $m^{-3}$) and non-control (5.3 µg $m^{-3}$) periods. This is consistent with the small differences in $SO_2$ concentrations for the NCCPC-control (8.5 µg $m^{-3}$, Figure S4) versus the non-control (12.4 µg $m^{-3}$, Figure S4) periods. Indeed, the low $SO_2$ concentrations may not have provided sufficient gaseous precursors to form substantial amounts of sulfate."

*3. Figures 4 and 5. Copy the source legend from Fig 5 to Fig 4.*

**Response:** Change made. Please see our revised Figure 4 in the revised manuscript.

*4. Page 10, lines 64 -72. First, as pointed earlier, the approach to determine the "stable conditions" is rough. Second, the samples (3 days and 2 days) for the stable conditions are too small, which would make the comparison statistically no meaningful. A better analysis is needed to separate the impacts of emission reduction and meteorological conditions.*

**Response:** As noted above, it has not been possible for us to obtain the emission inventory for the NCCPC control period. Therefore, our analysis of relatively stable atmospheric conditions was the best approach we had for evaluating the effectiveness of control measures. As the control measures were only in place for a short amount of time, this comparison is limited but it does support the argument that control measures were effective. We note in the revised manuscript that results of other studies also have shown short-term emission controls reduced pollutant levels, so our results were not unexpected. Following the reviewer's suggestion, we added more analysis of the meteorological conditions in the revised manuscript. Please see our response above.

*5. Page 15, lines 42-43. It is surprising that with an averaged surface concentration of 6.0ug/m3, EC imposes the largest cooling effects in surface DRF during the non-control period and several factors higher than that of OM, while the light extinction by OM is much higher than by EC. An explanation would be helpful.*

**Response:** The concentration of EC was 6.0 $\mu g/m^3$, and the light absorption caused by EC accounted for 14.3% of light extinction coefficient. The large contribution of EC absorption may be attributed enhancements caused by internal mixing with other materials because that process has been shown to amplify the light absorption of EC. In the revised manuscript, we added the following explanation: "The high EC DRF may have been due in part to EC particles internally mixed with other materials because mixing can amplify light absorption and thereby increase DRF."

*6. Fig 8 seems too small and a little bit complicated, which make it difficult to the reader to understand the effects of meteorological conditions on air quality in the BTH area. In addition, the location of the Xianghe site should be specified in the figure. Similar figure for October 12-23 might also be needed when you do analysis in decomposing the influences of the emission reduction and meteorology (especially for the five "stable" days).*

**Response:** Following the reviewer's suggestion, we modified the Figure 8 (also see Figure R3 below) in the revised manuscript. Moreover, the surface weather charts for the five "stable" days were added in the revised supporting information. Please see the response above.

[Figure]

**Figure R3.** Surface weather patterns at 08:00 (local time) over East Asia from 22 October to 2 November 2017. The black triangle represents Xianghe.

*Technical*
*The language need to be polished. The authors need go through the manuscript carefully and make edits. Following are just a few pickups.*

**Response:** The revised manuscript was polished by a native English speaker. Please see our new manuscript.

*Page 2 line 35, page 15 line 46, page 17 line 82: change "would" to "should"? Page 2 line 44, change "experienced" to "experiencing"?*

**Response:** Change made.

*Page 3 line 78, change "low-voltage" to "low-pressure".*

**Response:** Change made.

*Page 14 line 88, "genesis"?*

**Response:** In the revised manuscript, we revised our original expression to "The calculated mean bias and RMSE for $PM_{2.5}$ were -6.8 and 32.8 µg m$^{-3}$, and the index of agreement was 0.75, indicating that the formation of $PM_{2.5}$ during the two pollution episodes was reasonably well captured by the WRF-Chem model even though the predicted average $PM_{2.5}$ mass concentration of was lower than the observed value."

---

## Author Response (AR2)

Co-Editor Decision: Publish subject to minor revisions (review by editor) (26 Jan 2019) by Luisa Molina

Comments to the Author:

Dear Authors,

I am pleased to accept your manuscript for publication after considering an additional comment from one of the referees, "In principle, all factors identified in the PMF analysis need to be independent of each other. However in the manuscript the 3rd factor is a derivative of other 4 factors. A more detailed clarification and/or explanation is needed."

I would appreciate if you could respond to this comment. Best regards, Luisa Molina

**Response:** Thanks very much for the editor's kind decision. Following your suggestion, we changed the 3rd factor of "secondary particle formation" to "secondary source", which should be independent from other primary sources identified in the manuscript. The Figure 4 and 5 were also modified accordingly.

**Impacts of short-term mitigation measures on PM2.5 and radiative effects: case study at a regional background site near Beijing, China**

3 Qiyuan Wang1\*, Suixin Liu1, Nan Li2, Wenting Dai1, Yunfei Wu3, Jie Tian4, Yaqing Zhou1, Meng Wang1,

4 Steven Sai Hang Ho1, Yang Chen5, Renjian Zhang3, Shuyu Zhao1, Chongshu Zhu1, Yongming Han1,6,

5 Xuexi Tie1, Junji Cao1,7\*

[revised manuscript text omitted]

amounting to 36.7% of bext, and it was followed by OM (33.3%), EC (16.2%), (NH4)2SO4 (11.9%), and 519 fine soil (1.9%). The contributions of the various PM2.5 components to bext were different compared with 520 previous studies of the pollution controls for the Olympics and APEC. For example, Li et al. (2013) 521 reported that (NH4)2SO4 (41%) had the largest contribution to bext during the Olympics, followed by 522 523 NH4NO3 (23%), OM (17%), and EC (9%); Zhou et al. (2017) found that OM (49%) was the largest contributor to best during the APEC summit, followed by NH4NO3 (19%), (NH4)2SO4 (13%), and EC 524 (12%). These differences may be attributed to variable efficiencies of the controls for the specific fine 525 particle species and to variations in RH among studies, the latter of which can influence sulfate and nitrate 526 formation. 527

[revised manuscript text omitted]

Aerosol NO3- showed the largest contribution to PM2.5 mass (26.8%), and the high RH during PE1 likely promoted aqueous reactions involving nitrate. In contrast, OM (31.8%) was the most abundant species in PM2.5 during the PE2, and the PMF indicated that biomass burning was the largest source, accounting for 36.0% of the PM2.5 mass. The WRF-Chem simulation showed that the BTH region contributed 73.6% and 46.9% of PM2.5 mass during the PE1 and PE2, respectively. Calculations based on methods developed for the IMPROVE program indicated that OM was the largest

contributor (43.5%) to the chemical best during the non-control period, followed by NH4NO3 (32.4%), EC 592 (14.3%), (NH4)2SO4 (7.6%), and fine soil (2.2%). During the NCCPC-control period, NH4NO3 accounted 593 for 36.7% of bext, and that was followed by OM (33.3%), EC (16.2%), (NH4)2SO4 (11.9%), and fine soil 594 595 (1.9%). The TUV model showed that the estimated average DRF ( $-14.0 \pm 3.0$  W m-2) at the surface during the NCCPC-control period was 27.5% less negative than in the non-control period (-19.3  $\pm$  8.6 W m2), 596 597 and this is consistent with the lower PM2.5 loadings during the NCCPC-control period. Furthermore, EC 598 had the largest (most negative) influence on DRF at the surface during the non-control period; the EC DRF value of -13.4 W m-2 was followed by OM (-3.0 W m-2), NH4NO3 (-2.2 W m-2), (NH4)2SO4 (-0.5 W 599 m-2), and fine soil (-0.15 W m-2). The DRF values caused by EC, NH4NO3, OM, and fine soil when the 600 controls were in place were lower by -10.1, -1.7, -1.6, and -0.09 W m-2, respectively, compared with the 601 non-control period, and the corresponding percent reductions were 24.6, 22.7, 46.7, and 40.0%. The 602 603 results suggest that the short-term mitigation measures during the NCCPC-control period were effective in reducing fine particle pollution and those actions also had radiative effects sufficient to affect surface 604 605 temperature.

**606 Author contribution**

[revised manuscript text omitted]